# Development of an NLR-ID Toolkit and Identification of Novel Disease-Resistance Genes in Soybean

**DOI:** 10.3390/plants13050668

**Published:** 2024-02-28

**Authors:** Wei Shao, Gongfu Shi, Han Chu, Wenjia Du, Zikai Zhou, Hada Wuriyanghan

**Affiliations:** Key Laboratory of Forage and Endemic Crop Biotechnology, Ministry of Education, School of Life Sciences, Inner Mongolia University, Hohhot 010070, China; shaowei1995812@163.com (W.S.); y214575@126.com (G.S.); 13245118382@163.com (H.C.); 13754043691@163.com (W.D.); 67983286@163.com (Z.Z.)

**Keywords:** soybean, NLR, integrated domain, disease resistance, effector interaction, SRZ4

## Abstract

The recognition of pathogen effectors through the nucleotide-binding leucine-rich repeat receptor (NLR) family is an important component of plant immunity. In addition to typical domains such as TIR, CC, NBS, and LRR, NLR proteins also contain some atypical integrated domains (IDs), the roles of which are rarely investigated. Here, we carefully screened the soybean (*Glycine max*) genome and identified the IDs that appeared in the soybean TNL-like proteins. Our results show that multiple IDs (36) are widely present in soybean TNL-like proteins. A total of 27 *Gm-TNL-ID* genes (soybean TNL-like gene encoding ID) were cloned and their antiviral activity towards the soybean mosaic virus (SMV)/tobacco mosaic virus (TMV) was verified. Two resistance (*R*) genes, *SRA2* (SMV resistance gene contains AAA_22 domain) and *SRZ4* (SMV resistance gene contains zf-RVT domain), were identified to possess broad-spectrum resistance characteristics towards six viruses including SMV, TMV, plum pox virus (PPV), cabbage leaf curl virus (CaLCuV), barley stripe mosaic virus (BSMV), and tobacco rattle virus (TRV). The effects of Gm-TNL-ID^X^ (the domain of the *Gm-TNL-ID* gene after the TN domain) on the antiviral activity of a R protein SRC7^TN^ (we previously reported the TN domain of the soybean broad-spectrum resistance gene SRC7) were validated, and most of Gm-TNL-ID^X^ inhibits antiviral activity mediated by SRC7^TN^, possibly through intramolecular interactions. Yeast-two-hybrid (Y2H) and bimolecular fluorescence complementation (BiFC) assays showed that seven Gm-TNL-ID^X^ interacted with SMV-component proteins. Truncation analysis on a broad-spectrum antiviral protein SRZ4 indicated that SRZ4^TIR^ is sufficient to mediate antiviral activity against SMV. Soybean cDNA library screening on SRZ4 identified 48 interacting proteins. In summary, our results indicate that the integration of IDs in soybean is widespread and frequent. The NLR-ID toolkit we provide is expected to be valuable for elucidating the functions of atypical NLR proteins in the plant immune system and lay the foundation for the development of engineering NLR for plant-disease control in the future.

## 1. Introduction

Plants encounter the invasion of various pathogenic microorganisms in nature. During co-evolution with pathogens, host plants developed sophisticated immune systems [1]. Cell surface pattern-recognition receptors (PRRs) can recognize conserved microbial-associated molecular patterns (MAMPs), pathogen-associated molecular patterns (PAMPs), or host-derived damage-associated molecular patterns (DAMPs), thereby initiating the first layer of the immune response called pattern-triggered immunity (PTI) [2]. However, pathogens would evolve to evade PTI and infect the host via the effectors which are secreted into the extracellular vesicles or are transmitted to host cells. Plants produce a second layer of immune response that relies on R proteins to specifically recognize certain pathogen-derived effectors called avirulence (Avr) proteins, leading to the activation of a stronger immune response called effector-triggered immunity (ETI). ETI is often accompanied by local cell death in plants, known as a hypersensitivity response (HR) [3]. The largest group of R proteins identified to date are nucleotide-binding site-leucine rich repeat (NBS-LRR, NLR) proteins, which can recognize pathogen effectors directly or indirectly [4]. However, recent studies have also shown that PTI and ETI are interrelated and are sometimes difficult to distinguish as desired [5].

The plant NLR family belongs to the STAND (signal transduction ATPases with numerical domains) AAA+ superfamily. Typical plant NLRs have a multi-domain structure composed of a central NBS and a C-terminal LRR domain. Based on the structure of N-terminal domains, plant NLRs are usually divided into three categories: TIR-NBS-LRR (TNL), CC-NBS-LRR (CNL), and RWP8-NBS-LRR (RNL), each class possessing an N-terminal Toll/intelleukin-1 receptor (TIR), a coiled coil (CC), and resistance to powdery mildew 8 (RPW8)-like domains, respectively [6,7]. In addition to these conservative structural domains, some NLRs also include atypical IDs that exhibit polymorphic distributions. Most IDs resemble known effector target proteins, immune signal regulatory factors, or NLR cofactors, and are proposed to act as bait, directly binding pathogenic effectors or indirectly sensing effector activity [8,9,10].

The so-called “integrated decoy” model was proposed to explain the mechanism of ID activity, mostly based on research from the *RPS4*/*RRS1* gene pair [11] in *Arabidopsis thaliana* and the *RGA4*/*RGA5* gene pair [12] in rice. The ID is fused to a member of the NLR pair, acting as a bait and triggering the defense response of the second NLR upon binding the effector. In normal conditions, NLRs (sensors) with IDs are mainly responsible for inhibiting the activity of another NLR (executor). When pathogens invade, the sensor NLR recognizes the corresponding pathogen effector and loses inhibition on the executor, thereby mediating resistance. In addition, the two closely linked *NLR* genes, except for *RPP2A*/*RPP2B* [13], are transcribed in a relatively short intergenic region in a “head to head” orientation with a common promoter, suggesting their co-regulation at the transcriptional level [8]. For the *RPS4*/*RRS1* gene pair, the WRKY domain of RRS1 serves as a bait to bind to the pathogen effector PopP2, an acetyl-transferase from *Ralstonia solanacearum*, and AvrRps4 from *Pseudomonas syringae pv. pisi*, and trigger immune signaling mediated by its neighboring NLR RPS4 [14,15]. For the *RGA4*/*RGA5* gene pair, the HMA domain of RGA5 directly binds to the effectors AVR-Pia and AVR1-CO39 from rice blast fungus *Magnaporthe oryzae*, and activates RGA4-mediated resistance and cell death [16,17]. Multiple gene pairs that conform to the “integrated decoy” model have been identified to date, including some *TNL* pairs such as *Arabidopsis RRS1B*/*RPS4B* gene pairs [18], melon *Fom-1*/*Prv* gene pairs [19], and *Arabidopsis CHS3*/*CSA1* gene pairs [20], as well as some *CNL* pairs such as rice *Pik-1*/*Pik-2* alleles, *Pi5-1*/*Pi5-2*, *Pii-1*/*Pii-2*, and *Pias1*/*Pias2* gene pairs [21,22,23,24], wheat *Lr10*/*RGA2* gene pairs [25], barley *HvRga1*/*Rpg5* gene pairs [26], and rapeseed *BnRPR1*/*BnRPR2* gene pairs [27].

However, there are exceptions to the mode of above-mentioned ID activity, and there are also cases in which a single NLR protein with IDs can also function individually. For example, the N-terminus of the tomato immune protein Prf interacts with Pto kinase to regulate plant-specific immunity [28]. The wheat yellow–rust resistance-proteins Yr5, Yr7, and YrSP carry an N-terminal BED domain, which is necessary for their resistance [29]. The tomato spot wilt virus-resistance protein sw-5b has an N-terminal SD domain, which mediates resistance by changing the protein conformation in the presence or absence of pathogens [30]. The wheat stripe rust-resistance YrU1 protein contains both the N-terminal ankyrin repeat domain and the C-terminal WRKY domain, and this ankyrin repeat domain exhibits homologous oligomerization upon pathogen invasion [31]. The soybean NLR protein (GmNLR-ID#85) has a C-terminal WRKY domain which specifically recognizes the type III effector PopP2 from *Ralstonia pseudosolanacearum* [32].

In-depth research on NLR proteins and the mining of public genomic databases identified that IDs are commonly and frequently present in plant NLR proteins. Kroj et al. identified 162 putative IDs from 33 plant genomes using the GreenPhyl database, clarifying the high diversity of IDs [33]. Sarris et al. reported that 265 unique IDs from 37 terrestrial plant are present in plant NLRs [34]. Stein et al. identified highly variable structures in hundreds of different NLR-IDs proteins in 13 rice varieties [35]. The above studies indicate that the integration of IDs occurs frequently and repeatedly during genomic evolution. The above examples reveal a universal strategy for plants to detect specific pathogen effectors, to mediate signal transduction, or to regulate their own protein conformation by integrating bait into immune receptors (NLR proteins), demonstrating an effective monitoring mechanism for plants. Although NLR-IDs have been characterized in various plant families, little is known about their emergence and subsequent evolution, as well as about how NLR-IDs adapt to rapidly evolving pathogen effectors. Therefore, the ID can be considered a highly effective functional unit, and the functional characterization of NLR-ID genes in different species would be helpful for in-depth research on plant–disease interaction mechanisms, NLR protein evolution, broadening virulence targets, and the artificial design of NLR proteins.

Cultivated soybean (*Glycine max*) originated in China and was domesticated from wild soybean (*Glycine soja* Sieb. and Zucc.). With a history of over 5000 years of soybean cultivation in China, it has been widely spread to various parts of the world, providing humans with our main plant oil and protein resources [36]. In 2010, researchers used the whole-genome shotgun approach to sequence the entire genome of *Glycine max* var. Williams 82, and most of the genome sequences were assembled into 20 chromosomal-level pseudomolecules [37]. Over the next 10 years, researchers continuously performed resequencing and GWAS (Genome-Wide Association Studies) on wild and cultivated soybeans. The excavation of these genomic resources provides extremely important resources and platforms for soybean research, which will vigorously promote soybean molecular design and breeding, and help achieve the “green revolution” of soybeans [38,39,40,41,42,43].

However, soybean is vulnerable to a variety of pathogens in its life history, and soybean mosaic virus (SMV) is one of the main threats to the soybean industry [44,45]. SMV belongs to *potyviruses*, which have single-stranded sense RNA genomes of approximately 10 kilobases (kb), and encode 10 mature functional proteins, named P1, HC-Pro, P3, 6K1, CI, 6K2, NIa-Vpg, NIa-Pro, NIb, and CP [46]. Previously, we cloned a soybean atypical broad-spectrum resistance gene, *SRC7*, which encodes a BSP (basic secretory protein) domain at its C-terminus [47]. Currently, there are no other IDs or paired *NLR* genes reported in soybeans. Therefore, here we aimed to identify and functionally characterize the soybean *TNL-ID* genes (IDs located at the C-terminus) by the screening and cloning of the soybean *TNL-ID* gene and by verifying its viral immune activity, identifying the interaction between ID and SMV-component proteins, and characterizing SMV-resistance genes. Altogether, our research provides a valuable resource for the study of the *NLR-ID* gene resistance to pathogens and the diverse functions of its IDs.

## 2. Results

### 2.1. Screening of Soybean NLR Proteins with Putative IDs

While most of the disease R proteins comprise the characteristic NLR (NBS-LRR) structure, more and more NLR-like proteins with atypical domains have been identified to play important roles in plant immune responses [10]. Previously, we reported a soybean broad-spectrum resistance gene, *SRC7* (SMV resistance cluster 7), which encodes two characteristic domains, TIR and NBS, as well as an atypical domain BSP (basic secretory protein), among which the TIR-NBS region showed a basal resistance function while the C-terminal BSP domain was shown to be implicated in SMV recognition [47]. These studies suggest the role of atypical R proteins in plant immunity and prompt us to search for more atypical R proteins.

Currently, typical NLR proteins are generally classified into three classes, TNL, CNL and RNL, which harbors TIR or CC domains in the N-terminus, respectively. In addition, the NBS domain is always present in the atypical R proteins discovered so far, while the C-terminal LRR domain may have been replaced by other domains. In order to gain a deeper understanding of the evolution and diversity of the soybean TNL protein structure, as well as to identify atypical domains in TNL-like proteins, we used the SRC7^TIR^ and SRC7^TN^ sequence to screen throughout the soybean genome (Wm82. a2. v1, https://soybase.org/) (accessed on 15 December 2020) database using BLAST search. A total of 210 TNL-like proteins were retrieved from this screening (Appendix A). As shown in Appendix A, these genes are distributed on all 20 pairs of chromosomes in soybean, and are more densely distributed on chromosomes 2, 3, 6, 12, and 16.

In order to identify the domain structure of these TNL-like proteins, they were annotated in PFAM (http://pfam-legacy.xfam.org/) (accessed on 18 December 2020) and SMART (http://smart.embl-heidelberg.de/) (accessed on 18 December 2020) databases (Appendix A). As a result, a total of 36 different types of IDs were identified and were retrieved with different frequencies (Table 1).

We classified these TNL-like proteins based on their IDs (Appendix A). For simplicity, the domains other than TIR or TIR-NBS were referred to as the X domains. It should be noted that some of the C-terminal X domain includes both LRR and IDs. Based on this, these TNL-like proteins were divided into seven categories: T; TX; XT; TN; XTNL; TXNL; and TNX, with TNX accounting for the largest proportion. In addition, in the TNX class, proteins with only the LRR domain account for a large proportion, while the rest are R proteins with both ID and LRR domains (37%) (Appendix A). Next, we conducted polymorphism analysis on the identified IDs and found that they were distributed in different positions in soybean TNL proteins (Figure 1A). These IDs were further classified into three categories based on their predicted functions, namely nucleic acid binding, other signaling pathways, and DUF (domain of unknown function) (Figure 1B). In addition, we also performed an analysis on the frequency of these IDs, and found that the MotA_activ domain and zf-RVT domain have higher frequencies of occurrence (Figure 1C).

### 2.2. Analysis of the Antiviral Activity of 27 Gm-TNL-ID Genes against SMV/TMV

SMV causes a serious viral disease in soybean, and host genes conferring SMV resistance are mainly elusive. In order to identify disease-resistance genes from the above-identified *TNL-like* genes, we first tested their resistance to SMV. To visualize SMV infection to expedite our study, we used the SMV- GFP (green fluorescent protein) infectious clone developed in our lab [48]. In addition, due to the difficulty of transient infection and phenotype observation in soybean, we used *N. benthamiana* to express these genes. As some of these genes share homology with the tobacco *N* gene (a TNL-type TMV-resistance gene [49]), the TMV-GFP infectious clone was also used in this study [50].

As a comparison to the domain structure of SRC7 (SRC7 has typical TIR and NBS domains, lacking LRR domains and having a BSP domain at the C-terminus) (Appendix A), we ultimately screened 41 *TNL-like* genes (these genes all encode typical TIR and NBS domains, with IDs encoded after the TN domain) and named them *Gm-TNL-ID*, involving a total of 23 IDs (Appendix A). Among them, apart from *KR3* (*Gm-TNL-ID25*, *GLYMA_06G267300*) [51], *KR1* (*Gm-TNL-ID29*, *GLYMA_19G054900*) [52], and *GLYMA_16G135500* (*Gm-TNL-ID35*) [53], and a recently reported soybean *NLR* gene *GmNLR-ID#85* (*Gm-TNL-ID17*, *GLYMA_05G165800*) [32], no reports are seen for the others to date. Then, the open reading frame (ORF) sequences of these genes were cloned into the binary vector pCambia1300 under the constitutive 35S promoter, and their roles in SMV and TMV resistance were investigated using the *N. benthamiana* transient expression system. In total, 27 *Gm-TNL-ID* genes were successfully cloned in this study.

Soybean SRC7 and SRC7^TN^ were used as positive controls in an antiviral assay against SMV/TMV, as the TIR-NBS domain of SRC7 (SRC7^TN^) showed resistance against these two viruses. Eleven out of twenty-seven candidate *Gm-TNL-ID* genes showed resistance towards SMV/TMV, among which two genes, *Gm-TNL-ID5* (*GLYMA_15G152400*) and *Gm-TNL-ID14* (*GLYMA_16G210600*), showed full resistance, while the other nine genes including *Gm-TNL-ID2* (*GLYMA_06G265000*), *Gm-TNL-ID10* (*GLYMA_09G056400*), *Gm-TNL-ID12* (*GLYMA_16G127900*), *Gm-TNL-ID16* (*GLYMA_06G285500*), *Gm-TNL-ID17*, *Gm-TNL-ID23* (*GLYMA_16G006400*), *Gm-TNL-ID26* (*GLYMA_16G213700*), *Gm-TNL-ID29*, and *Gm-TNL-ID39* (*GLYMA_14G151100*) showed partial resistance (Figure 2A,B). To confirm this, we used software Image J v1.8.0 to quantify the fluorescence diffusion in infected leaves. The results showed that the fluorescence intensity of nine partial resistance genes was significantly reduced compared to that of the EV control, indicating their antivirus activity on SMV/TMV (Figure 2C,D). In addition to fluorescence detection, we further confirmed the antiviral activity of these 27 *Gm-TNL-ID* genes against SMV and TMV by detecting viral titers via a real-time quantitative PCR (RT-qPCR) technique. As a result, the two genes, *Gm-TNL-ID5* and *Gm-TNL-ID14*, caused 68% and 59% decreases in SMV viral accumulation, 83% and 48% decreases in TMV viral accumulation, and the other nine partially resistant genes caused 22–38% decreases in SMV virus accumulation, and 26–37% decreases in TMV virus accumulation (Figure 2E,F). This is consistent with the phenotypic data obtained above. As the genes *Gm-TNL-ID5* and *Gm-TNL-ID14* encode the proteins carrying the AAA_22 and zf-RVT integrated domain and both genes have complete resistance to SMV, furthermore, two genes containing AAA_22 domains were cloned in this study (*Gm-TNL-ID4* (*GLYMA_12G240100*) and *Gm-TNL-ID5*), five genes containing zf-RVT domains were cloned in this study (*Gm-TNL-ID11* (*GLYMA_16G210800*), *Gm-TNL-ID12*, *Gm-TNL-ID13* (*GLYMA_09G161400*), *Gm-TNL-ID14*, and *Gm-TNL-ID15* (*GLYMA_08G301200*)). We named them *SRA2* (SMV-resistance gene contains the AAA_22 domain) and *SRZ4* (SMV-resistance gene contains the zf-RVT domain), respectively.

### 2.3. Examination of Broad-Spectrum Antiviral Resistance of 27 Gm-TNL-ID Genes

Plant viral diseases are considered the main factors restricting crop production. Therefore, broad-spectrum and persistent resistance are indispensable for crop improvement. Broad-spectrum resistance refers to the resistance of a cultivar or a gene to two or more pathogens or the majority of races/isolates of the same pathogen [54,55]. Soybean is an important economic oil crop, but it is susceptible to various viruses [56]. Therefore, the mining of broad-spectrum resistance genes is of great significance.

In the above experiment, the antiviral activity of 27 *Gm-TNL-ID* genes on SMV/TMV (*Tobamovirus*) was assayed. Next, we asked if these *Gm-TNL-ID* genes were resistant to four other viruses, including PPV (*Potyvirus*) from the same potato Y virus genus with SMV, CaLCuV (*Begomovirus*), which has a wide host range, TRV (*Tobravirus*), and BSMV (*Hordeivirus*). As shown above, the *Gm-TNL-ID* genes were transiently expressed in *N. Benthamiana*, which were subsequently infiltrated by the four viruses, respectively, and the resistance level was examined through viral titers using a RT-qPCR approach.

As a result, ten genes including *Gm-TNL-ID1* (*GLYMA_13G194700*), *SRA2*, *Gm-TNL-ID8* (*GLYMA_13G078200*), *Gm-TNL-ID11*, *SRZ4*, *Gm-TNL-ID15*, *Gm-TNL-ID19* (*GLYMA_03G077400*), *Gm-TNL-ID30* (*GLYMA_19G054700*), *Gm-TNL-ID35*, and *Gm-TNL-ID36* (*GLYMA_16G135200*) caused 18–74% decreases in PPV viral accumulation (Figure 3A). Fifteen genes including *Gm-TNL-ID1*, *Gm-TNL-ID3* (*GLYMA_03G087500*), *SRA2*, *Gm-TNL-ID7* (*GLYMA_16G087100*), *Gm-TNL-ID8*, *Gm-TNL-ID9* (*GLYMA_06G310000*), *Gm-TNL-ID10*, *Gm-TNL-ID12*, *SRZ4*, *Gm-TNL-ID18* (*GLYMA_03G075300*), *Gm-TNL-ID25*, *Gm-TNL-ID29*, *Gm-TNL-ID35*, *Gm-TNL-ID37* (*GLYMA_03G052800*), and *Gm-TNL-ID39* caused 27–90% decreases in CaLCuV viral accumulation (Figure 3B). Five genes including *SRA2*, *SRZ4*, *Gm-TNL-ID16*, *Gm-TNL-ID23*, and *Gm-TNL-ID26* caused 23–90% decreases in BSMV viral accumulation (Figure 3C). Eleven genes including *SRA2*, *Gm-TNL-ID10*, *SRZ4*, *Gm-TNL-ID15*, *Gm-TNL-ID16*, *Gm-TNL-ID17*, *Gm-TNL-ID21* (*GLYMA_16G085700*), *Gm-TNL-ID30*, *Gm-TNL-ID36*, *Gm-TNL-ID37*, and *Gm-TNL-ID39* caused 34–85% decreases in TRV viral accumulation (Figure 3D). Among these genes, the *SRA2* and *SRZ4* genes showed significant resistance to PPV, CaLCuV, BSMV, and TRV. Taken together, *SRA2* and *SRZ4* exhibit significant resistance to at least six viruses (SMV, TMV, PPV, CaLCuV, BSMV, and TRV). Therefore, these two genes are selected as broad-spectrum viral resistance genes and will provide valuable resources for the breeding of plant-resistant varieties.

### 2.4. Effect of X Domain from Gm-TNL-ID Genes on Antiviral Resistance of SRC7^TN^

In our previous study, SRC7^TN^ showed significant resistance to SMV/TMV, which was evidenced by the appearance of the HR phenotype, while the BSP domain of SRC7 might affect the protein stability or mediate viral recognition [47]. The 41 Gm-TNL-ID proteins mentioned above all have a relatively complex C-terminal domain (LRR + ID). In view of this, we named the C-terminal domain of these genes after TN as X domains and used SRC7^TN^ to preliminarily verify the function of these X domains. Therefore, we placed the X domains C-terminally to SRC7^TN^ in-frame through homologous recombination, and ultimately obtained 32 SRC7^TN^-ID^X^ constructs. Next, the antiviral activity of them on SMV/TMV was verified through tobacco transient expression experiments. SRC7^TN^-ID9^X^, SRC7^TN^-ID18^X^, SRC7^TN^-ID25^X^, and SRC7^TN^-ID36^X^ showed similar resistance compared with SRC7^TN^, indicating that they have no significant effects on SRC7^TN^. In the remaining 28 SRC7^TN^-ID^X^ groups, the plants all showed varying intensities of fluorescence (viral appearance), indicating that these 28 ID^X^ domains had varying degrees of inhibitory effect on SRC7^TN^ resistance for SMV/TMV (Figure 4A,B).

In order to investigate how these ID^X^ domains affect SRC7^TN^-mediated SMV/TMV resistance, we used a Y2H system to examine whether there is interaction between these ID^X^ domains and SRC7^TN^ domains. First, we performed an auto-activation assay on these X domains in the Y2H system. Seven of them, Gm-TNL-ID8^X^, Gm-TNL-ID10^X^, Gm-TNL-ID15^X^, Gm-TNL-ID24^X^, Gm-TNL-ID26^X^, Gm-TNL-ID37^X^, and Gm-TNL-ID38^X^, exhibited strong auto-activation (Appendix A), and were excluded from following interaction assay with SRC7^TN^. The remaining 25 Gm-TNL-ID^X^ were rendered to following assays.

In the Y2H experiment, The combination of 8 Gm-TNL-ID^X^ domains and SRC7^TN^ showed good growth and a blue color on the yeast selective media (SD/-L/-W/-H/-A), while the combination of 14 Gm-TNL-ID^X^ domains and SRC7^TN^ showed good growth and a blue color on yeast selective media (SD/-L/-W/-H), indicating 22 Gm-TNL-ID^X^ domains showed interactions with SRC7^TN^, while three ID^X^ (Gm-TNL-ID3^X^, Gm-TNL-ID16^X^, Gm-TNL-ID21^X^) domains showed no interaction with SRC7^TN^ (Figure 4C), indicating that they might influence SRC7^TN^-mediated antiviral activity via direct intramolecular domain interaction. However, it is worth noting that Gm-TNL-ID3^X^, Gm-TNL-ID16^X^, and Gm-TNL-ID21^X^ all exhibit inhibitions of SMV/TMV resistance mediated by SRC7^TN^. We speculate that this may be due to the complex C-terminal structure disrupting protein conformation, thereby affecting protein function.

### 2.5. Screening of Interaction between the Gm-TNL-ID^X^ Domain and SMV Component Proteins

The LRR domain of plant NLR proteins is typically involved in the recognition of pathogen effectors, leading to the oligomerization of NLR proteins and the formation of larger NLR protein-containing complexes called resistosomes [57,58]. During evolution, the NLR protein possessed diverse functions by acquiring additional IDs that are typically involved in the recognition of pathogen effectors, resistance signal transduction or maintaining stable protein conformation, and some *NLR* genes with IDs may act as sensors and may not exhibit resistance themselves [8,28,29,30,31,59].

Therefore, considering the potential role of LRR and IDs in pathogen recognition, we screened the 25 Gm-TNL-ID^X^ domains through Y2H system to determine whether there is any interaction between them and the 10 component proteins encoded by the SMV genome. As shown in Figure 5A, Gm-TNL-ID3^X^- and SMV-component proteins CP, P3, and 6K2 showed interactions; Gm-TNL-ID13^X^ and 6k2 showed interactions; Gm-TNL-ID16^X^ and P1, NIb showed interactions; Gm-TNL-ID25^X^ and NIb, HC-Pro showed interactions; Gm-TNL-ID29^X^ and 6K2, NIb showed interactions; Gm-TNL-ID30^X^ and HC-Pro showed interactions; and Gm-TNL-ID35^X^ and P1, P3, and 6K2 showed interactions. In BiFC assays using *N. benthamiana* plants, we further validated the interaction between the seven Gm-TNL-ID^X^ domains screened above and SMV-component proteins. The results showed that the combinations of seven Gm-TNL-ID^X^ domains and SMV-component proteins screened through the Y2H system all exhibited yellow fluorescence on the cell membrane of tobacco leaves (Figure 5B). The results further demonstrate the interaction between these seven Gm-TNL-ID^X^ domains and corresponding SMV-component proteins. The above interaction results are summarized in Table 2. The results indicate that at least some, if not all, of these Gm-TNL-ID^X^ domains are involved in SMV recognition/resistance by mediating R–effector interactions.

### 2.6. Further Analysis of Antiviral Mechanism of SRZ4

Given the broad-spectrum antiviral activity of *SRZ4* and its novel zf-RVT and RVT3 IDs, we seek to further explore the resistance mechanism of *SRZ4* and the function of its ID. When the SRZ4 was fused with GFP and transiently expressed in *N. benthamiana* leaf epidermis cells, a clear membrane localization of SRZ4: GFP was observed (Figure 6A), indicating its function as membrane receptor. Based on the domain structure of SRZ4 (Appendix A), we designed and made seven SRZ4 truncations, including SRZ4^TIR^, SRZ4^NBS^, SRZ4^TN^ (abbreviation for TIR-NBS [TN] of SRZ4), SRZ4^TNL^ (abbreviation for TIR-NBS-LRR [TNL] of SRZ4), SRZ4^TNLZ^ (abbreviation for TIR-NBS-LRR-zf-RVT [TNLZ] of SRZ4), SRZ4^NLZ3^ (abbreviation for NBS-LRR-zf-RVT-RVT3 [NLZ3] of SRZ4), and SRZ4^Z3^ (abbreviation for zf-RVT-RVT3 [Z3] of SRZ4). Truncation analysis showed that SRZ4^TIR^ was sufficient to induce complete resistance to SMV/TMV. Truncation with a TIR domain showed resistance, while truncation without a TIR domain did not show resistance. The presence or absence of an ID did not show a significant difference in its resistance. The above results indicate that the TIR domain is essential and sufficient for the antiviral activity of *SRZ4* and determines the antiviral activity of the *SRZ4* gene (Figure 6B).

In order to further explore the resistance mechanism of SRZ4, Y2H assay was conducted using a soybean cDNA library via the bait SRZ4 to identify SRZ4-interactors from soybean. Yeast cells transformed with BD-SRZ4 could grow on SD/-W-minimal media plates but did not grow on SD/-W/-H-selective media plates. Yeast cells co-transformed with BD-SRZ4 and AD could grow on SD/-L/-W-minimal media plates not grow on SD/-L/-W/-H-selective media plates, indicating no toxicity and the auto-activation of SRZ4 (Appendix A). Therefore, the BD-SRZ4 construct is suitable for cDNA library screening.

After three generations of subculture, 89 blue colonies with good growth status were obtained on SD/-L/-W/-H/-A/X-α-Gal plates (Figure 7A). Then, the yeast colonies were PCR-amplified using sequencing primers (F: 3′AD/R: T7) (Figure 7B), and the colonies with different sizes and single bands were selected for sequencing analysis. After NCBI blast analysis of the sequencing results and excluding duplicate sequences, a total of 48 candidate proteins interacting with SRZ4 were preliminarily obtained (Appendix A). Domain prediction analysis was conducted on these proteins (Appendix A), followed by gene ontology (GO) enrichment analysis to annotate and classify the proteins according to their biological process, cellular component, and molecular function (Figure 7C). Among the biological process group, the interacting proteins were enriched in processes such as fatty acid transport, modification-dependent protein degradation metabolism, messenger ribonucleic acid splicing, photosynthesis, and protein ubiquitination. In the cellular component group, the interacting proteins were mostly distributed in membranes, chloroplast thylakoid membranes, cytoplasm, nucleus, and photosystem II. In the molecular function group, interacting proteins were enriched in functions such as chlorophyll binding, messenger ribonucleic acid binding, and structural constituents of ribosomes as well as ubiquitin protein–ligase binding.

## 3. Discussion

Plants are constantly engaged in an evolutionary arms race with pathogens. Although many NLR proteins use C-terminal LRRs to detect pathogen effectors, some of them have already obtained IDs that are involved in pathogen recognition or downstream signaling [60]. Kroj et al. investigated the distribution of IDs in NLR proteins from 31 plant species and found that these IDs appear frequently with an irregular distribution pattern. Furthermore, IDs appear at different positions in NLR proteins, and there are different IDs or multiple occurrences of the same ID in a single NLR protein [33]. For example, the rice *RGA5* gene encodes the HMA domain at its C-terminus [12]; the tomato *sw-5b* gene has an N-terminal SD domain [30]; the wheat *YrU1* gene contains both the N-terminal ankyrin repeat domain and the C-terminal WRKY domain [31]; and the *SRZ4* gene we screened encodes two duplicated RVT3 domains. Although the occurrence of NLR-ID is low in some plant species, its widespread distribution indicates that NLR diversification is often used by plants to expand their pathogen-recognition ability, enabling them to cope with rapidly evolving pathogen-derived molecules [33,34]. The main purpose of this study is to identify the atypical *TNL* genes in soybeans and evaluate the IDs that are encoded by them. We identified a total of 210 *TNL-like* genes in soybeans, including 58 *TNL-ID* genes and 36 different IDs, which can serve as an important reference for the diversification of NLR functions.

The functional model of typical NLR proteins is often associated with the conformational changes which are triggered by LRR binding to an effector, causing the NBS domain to exchange ADP for ATP and to reshape the N-terminal domain to initiate signal transduction [4,61]. Homology assessment indicates that the IDs present in NLR proteins may originate from functional *non-NLR* genes through repetitive events [59]. The most common IDs found in NLR proteins include WKRY, BED (BEAF and DREF proteins from *Drosophila*), zinc finger (Znf-BED) DNA binding domains, protein kinase domains, and other signaling molecule domains [33,34]. The function of the WRKY domain as bait has been well characterized in RRS1-R [14,15]. The Xa1 resistance protein from *O. sativa* possesses a BED domain, which recognizes the multiple transcription activator like effects of the bacterial blight pathogen *Xanthomonas oryzae* [62,63]. The wheat yellow rust resistance proteins Yr5, Yr7, and YrSP all encode the BED domain, which are necessary for pathogen response although their mechanism of action is not yet clear [29]. In addition, Kroj et al. showed that the BED domains of ZBED proteins in rice are related to its resistance toward *M. oryzae* [33]. A protein kinase domain in barley Rpg5 and wheat Tsn1 mediates the recognition of ToxA effector of necrotrophic fungi *Pyrenophora tritici repentis* and *Stagonospora nodorum* [26,64]. It is worth noting that some functional NLR proteins with kinase domains or DUF676 but lacking CC or TIR domains have also been characterized, indicating that such IDs can substitute the signaling function of these missing domains [65]. Here, we identified 36 IDs in soybean-atypical TNL proteins, more than half of which contained nucleic acid-binding domains such as RVT3 and MotA_activ. Additionally, a Gm-TNL-ID17 protein with a WRKY domain has recently been reported [32]. It seems that the IDs in plant NLR proteins have plural functions. Therefore, the functional characterization of these IDs will provide new insights into plant immunity and disease resistance.

Previously, we cloned a soybean broad-spectrum resistance protein SRC7 with the structure of TIR-NBS-BSP, among which SRC7^TN^ performed the resistance function while its atypical domain BSP appeared to dampen the resistance and mediate pathogen recognition [47]. Based on this, here we recombined the 32 Gm-TNL-ID^X^ into SRC7^TN^ and validated their function. The results showed that most of the Gm-TNL-ID^X^ (28) played a similar function to BSP, inhibiting the antiviral activity of SRC7^TN^ for SMV/TMV. In addition, most of these Gm-TNL-ID^X^ (25) interacted with SRC7^TN^ in Y2H assay. It has been well characterized that LRR domains play positive or negative regulatory roles in NLR proteins through intra- or inter-molecular interactions [66]. Therefore, we speculate that these ID^X^ may influence the antiviral activity of SRC7^TN^ through intramolecular interactions. However, the exact roles of these Gm-TNL-ID^X^ await further characterization in chimeric SRC7^TN^-ID^X^ and especially in their original protein backgrounds.

Given the above considerations, we cloned a total of 27 full-length *Gm-TNL-ID* genes and first verified their antiviral activity toward SMV and TMV. Eleven *Gm-TNL-ID* genes were shown to be SMV/TMV resistance genes, including two broad-spectrum resistance genes (*SRA2* and *SRZ4*) for six plant viruses in four genera, which provide valuable resources for viral resistance breeding. On the other hand, the main purpose of this study is to provide a toolkit for the identification of interactions between soybean *TNL-ID* genes and SMV effectors. Therefore, we performed a Y2H assay between 32 ID^X^ with 10 component proteins of SMV, and ultimately identified 7 ID^X^ which showed interactions with different component proteins of SMV. The strategy of using the ID^X^ to screen SMV effectors partially explains their function for viral recognition, and further detailed study is needed. The well-accepted “integrated decoy” model proposes that plant NLRs can be divided into sensor NLRs and helper NLRs, which are responsible for identifying pathogen effectors and immune signal outputs, respectively [67]. We speculate that these Gm-TNL-ID proteins with SMV recognition activity may play the role of sensor NLRs.

In addition, among the seven Gm-TNL-ID^X^ identified to interact with SMV component proteins, Gm-TNL-ID25^X^, Gm-TNL-ID29^X^, Gm-TNL-ID30^X^, and Gm-TNL-ID35^X^ all encode a MotA_activ domain. The transcription factors MotA, T4 bacteriophage co-activator AsiA, and *E. coli* RNA polymerase containing sigma-70 are necessary for activating the intermediate promoter of T4 bacteriophage. The MotA_activ domain family are often associated with transcriptional activation [68,69]. However, currently, no functional MotA transcription factors and MotA_activ domain-related proteins have been identified in plants. In rice, RGA5 was shown to identify the avirulence proteins AVR-Pia and AVR1-CO39 from the rice blast fungus *Magnaporthe oryzae* through the integrated heavy metal-associated (HMA) domain [17,70]. The sequence diversity of the HMA domain in the *Pik-1* allele enables it to recognize different AVR-Pik variants, indicating the fitness of the polymorphisms [71,72]. The *Pik-1* alleles Piks-1 and Pikm-1 only have two amino acid differences in the HMA domain but show recognition specificity [73,74]. Therefore, recognition specificity is likely related to the co-evolutionary dynamics between *Magnaporthe oryzae* and rice at the molecular level through direct protein–protein interactions in effector–HMA binding affinity [75,76]. It is worth noting that Gm-TNL-ID25^X^ includes a shorter C-end region and only one MotA_activ domain but does not contain any other LRR domains. These four MotA_activ domains show approximately 60% sequence similarity (Appendix A). We speculate that this sequence diversity determines recognition specificity.

Currently, multiple *NLR* genes have been reported to be implicated in pathogen responses in soybean, but there are few functional studies on the *NLR-ID* gene. In consideration of this, we focused our attention on *SRZ4* from the above-mentioned *Gm-TNL-IDs* for functional study. SRZ4 encodes a protein of TIR-NBS-LRR-zf-RVT-RVT3-RVT3 with three IDs and exhibits complete resistance to SMV/TMV. The RVT domain is a reverse transcriptase domain and the zf-RVT domain possesses a zinc finger structure. While reverse transcriptase was well characterized in RNA viruses [77], its involvement in NLR functions has not been reported yet. In this study, a total of four NLR-ID proteins with the RVT domain were cloned and functionally characterized. Among them, the *Gm-TNL-ID12* gene showed partial resistance to SMV/TMV, while *SRZ4* showed complete resistance to SMV/TMV. Interestingly, *Gm-TNL-ID12* and *SRZ4* showed higher sequence similarity compared with other NLR-ID proteins with the RVT domain (Appendix A). Although SRZ4 has antiviral activity, its C-terminal LRR domain and integrated domain SRZ4^X^ have not been identified to interact with the component proteins of SMV. Truncation analysis on SRZ4 indicates that the TIR domain is a key domain mediating SRZ4 resistance. At present, studies have shown that the TIR domain in plant NLR proteins plays multiple functions. The TIR domain is usually activated through self-association and has enzymatic activity. It can use NAD^+^ (Nicotinamide Adenine Dinucleotide+) or nucleic acid (RNA/DNA) as substrates to produce different nucleotide products as downstream signaling molecules [78,79]. Therefore, we speculate that SRZ4^TIR^ may also mediate plant disease resistance through its oligomerization, consistent with the conserved role of TIR domains in other TNL proteins.

NLR proteins are key players in plant immunity and are widely used in crop breeding programs. On the contrary, plant pathogens can produce multiple effectors, which evolve rapidly and form an arms race with plant NLR proteins. Therefore, the engineering of plant immune systems by expanding the ability of NLR to recognize a wider range of pathogens is important for improving crop disease resistance [80,81]. The *NLR-ID* gene has become a key target for NLR engineering due to its ability to recognize various pathogen effectors via their varying IDs. While the HMA domain represents the model ID in plant immunity [82,83] and has been engineered to mediate broad pathogen resistance [84,85,86], the mining and detailed analysis of other IDs in atypical NLR proteins could help us to design more specialized NLRs to cope with rapidly evolving pathogens. Therefore, our study provides a toolkit for the functional study of the IDs in plant NLR proteins, and the broad-spectrum antiviral gene SRZ4 reported here might have significant application value.

## 4. Materials and Methods

### 4.1. Whole Genome Sequence Analysis, Gene Screening, and Domain Prediction

We used SRC7^TIR^ and SRC7^TN^ protein sequences to perform BLAST screening in the soybean genome database (Wm82. a2. v1, https://soybase.org/) (accessed on 15 December 2020), respectively, and then removed duplicate sequences, and ultimately obtained 210 TNL-like candidate protein sequences. Chromosomal location of the candidate protein on 20 chromosomes of soybean were visualized using Mapgene2chrom website (http://mg2c.iask.in/mg2c_v2.0/) (accessed on 18 December 2020). Quantitative statistics and classifications of these proteins are based on differences in domain structure.

### 4.2. Bioinformatic Analysis

The candidate protein sequences and CDS sequences were obtained using the Ensemble Plants database (https://plants.ensembl.org/index.html) (accessed on 20 December 2020). We used PFAM database (http://pfam-legacy.xfam.org/) (accessed on 18 December 2020) and SMART database (http://smart.embl-heidelberg.de/) (accessed on 18 December 2020) to predict the structural domains and to calibrate the positions of these candidate proteins. All sequence alignments were performed using software GENEIOUS v4.8.4. Subcellular localization prediction was performed using the WoLFPSORT server (https://wolfpsort.hgc.jp) (accessed on 10 July 2022). The GO enrichment visualization analysis of the screened interaction proteins of SRZ4 were performed in the bioinformatics online website (https://www.bioinformatics.com.cn/) (accessed on 25 September 2022).

### 4.3. Plant Materials, Gene Cloning, Vector Construction and Soybean cDNA Library

All plant materials were grown in a standard greenhouse (temperature 25 °C, 16 h of light: 8 h of darkness, humidity: 60%). *Nicotiana benthamiana* 17 Wt strain was used for transient expression and protein localization assays at 3–4 weeks of age. Soybean Williams cultivars were used in this study. Total RNA was extracted from 6–8-week-old soybean leaves using Ttizol reagent (YESEN, cat: 10606ES60). cDNA was synthesized by the HiScript II 1st Strand cDNA Synthesis Kit (Vazyme, cat: R212.) The open reading frame (ORF) sequences of candidate genes were amplified from soybean *Williams* cultivar cDNA, and PCR products were cloned into pCE2 TA/Blunt-Zero Cloning Kit (Vazyme, cat: C601) and verified by sequencing. For the overexpression of candidate genes, the ORF sequence was cloned into the *Sac*I/*Kpn*I sites of binary vector pCambia1300 under the cauliflower mosaic virus (CaMV) 35S promoter. The recombinant vector was verified by sequencing and was transformed to *Agrobacterium tumefaciens*. All *Agrobacterium* strains used in this study were GV3101, except for the *Agrobacterium* strain EHA105 used in BSMV-related vector. For Y2H assay, the sequence was cloned into the *BamH*I/*Sal*I sites of binary vector pGBKT7 and *BamH*I/*EcoR*I sites of binary vector pGADT7. For BiFC assay, The ID^X^ sequence and SMV component proteins were cloned into the *EcoR*Ⅴsites of entry vector pQBV3, and subsequently inserted into the destination vector pDEST-vYNE (R)^GW^ or pDEST-vYCE (R)^GW^, in frame with nVenus or cVenus fragment. For subcellular localization, full-length *SRZ4* was amplified and ligated into pQBV3 and subsequently inserted into pEarleyGate103-SL vector. The soybean cDNA library was constructed by Shanghai OE Biotech Co., Ltd. (Shanghai, China), and experimental steps for screening cDNA library were performed according to the manufacturer’s instructions (https://www.oebiotech.com/wenkufuwu/nlibrary) (accessed on 25 July 2022). All of the primers used in this manuscript were provided in Appendix A.

### 4.4. Transient Expression of Selected Genes in Nicotiana Benthamiana

*Agrobacterium* strain-carrying recombinant vector was cultured in LB solid medium with appropriate antibiotics at 28 °C for 24 h, and then cultured in liquid medium with a ratio (50:1) for 4–6 h. Cell cultures were centrifuged and resuspended in infiltration buffer (10 mM MES, 10 mM MgCl_2_, 200 μM acetosyringone) and adjusted to an appropriate OD_600_ for infiltration. *Agrobacterium* carrying virus infectious clones (SMV-GFP or TMV-GFP) were inoculated in the same way. After incubating the mixed liquid (Gene final OD_600_: 0.8, viral infectious clone final OD_600_: 0.4) at room temperature and in the dark for 1–3 h, *N. benthamiana* leaves were infiltrated with a 1 mL syringe and cultivated in a greenhouse. Each experiment underwent three biological replicates, and each set of biological replicates included at least 20–30 technical replicates. After 4–5 dpi, GFP fluorescence was detected by a hand-held long-wave (365 nm) UV lamp. For validation of broad-spectrum resistance, the infectious clones of PPV, TRV, CaLCuV, and BSMV carried by *Agrobacterium* were transiently co-infiltrated with candidate genes in *N. benthamiana*. After 4 dpi, the infected leaf tissue was taken for virus titer detection.

### 4.5. Yeast-Two-Hybrid (Y2H) Assay

Different truncations of candidate genes were subcloned and inserted into the AD or BD vectors, respectively. In our pervious study, ORF sequences for SMV 10 component proteins were PCR amplified from the SMV genomic RNA and inserted into AD vector [47]. Yeast transformation with the Y2Hgold yeast strain and yeast growth assays were performed as described in the literature [47]. Briefly, the two recombinant vectors in equal proportion (AD:BD = 1:1100 ng) were transformed into Y2H competent cells, and were cultured at 30 °C for 3 days using minimal media (SD/-L/-W). Positive yeast transformants were diluted with gradients of 10^−1^, 10^−2^ and 10^−3^, and 2 μL of each colony was taken onto stringent selective media (SD/-L/-W/-H) and media (SD/-L/-W/-H/-A). AD-T/BD-lam transformants served as negative controls, AD-T/BD-53 transformants served as positive controls, and X-α-Gal was added to visualize the reporter expression. The plates were incubated at 30 °C for 4–5 days, and the well-grown colonies were photographed.

### 4.6. Bimolecular Fluorescence Complementation (BiFC) Assays and Determination of Protein Localization

BiFC assays were conducted as described in the literature [47]. In short, *Agrobacterium* carrying different nVenus and cVenus pairs were infiltrated into *N. benthamiana* leaves, and the Venus fluorescence signals were examined at 48–72 hpi using confocal laser scanning microscopy (Carl Zeiss AG, Oberkochen, Germany, LSM710). For subcellular localization, *Agrobacterium* strain-carrying recombinant vector and membrane maker (mCherry) were co-infiltrated into *N. benthamiana* leaves, and GFP signal was visualized at 48–72 hpi using confocal laser scanning microscopy.

### 4.7. Quantification and Statistical Analysis

For fluorescence quantification of tobacco leaves, the relative intensities of SMV-GFP/TMV-GFP were processed and quantified with software Image J v1.8.0 (National Institutes of Health, Bethesda, MD, USA). The fluorescence intensity of each gene was subjected to 10 technical replicates. For virus titer detection, total RNA was isolated from mixed samples of corresponding infection areas of 10 individual leaves and first-strand cDNA was synthesized after co-infiltrating tobacco leaves with different viral infectious clones and candidate genes for 4 days. The control group (EV) used pCambia1300 empty vector and corresponding viral infectious clones for mixed infection, and their antiviral activity was determined by comparing the accumulation level of the virus after adding the genes we screened. RT-qPCR was performed using the Applied Biosystems ViiA 7 using PerfectStart Green qPCR SuperMix Kit (TransGen Biotech, Beijing, China, TG-AQ601) following the manufacturer’s instructions. Three biological replicates were performed for each experiment. Graphs were generated by GraphPad Prism 8 (La Jolla, CA, USA), Excel 2019 (Microsoft, Redmond, WA, USA). Error bars represent the SD of the mean, and significance was indicated when *p* < 0.05.

## 5. Conclusions

In the present study, we comprehensively screened the IDs present in soybean *TNL-like* genes and validated the immune activity of 27 *Gm-TNL-ID* genes against various pathogens, identifying two broad-spectrum resistance genes *SRA2* and *SRZ4*. We used SRC^TN^ to preliminarily characterize the role of X domains (LRR + ID) located at the C-terminus of the gene, indicating that most of them have inhibitory antiviral activity. In addition, we screened for interaction effector factors between the Gm-TNL-ID^X^ domain and SMV component proteins, and identified seven ID^X^ domains that interact with SMV component proteins. We further selected the *SRZ4* gene for resistance truncation analysis and interaction protein screening, laying the foundation for the analysis of the *SRZ4*-resistance mechanism. Overall, the toolkit we provide for soybean *TNL-ID* gene research has important reference value for ID function analysis and the engineering of *NLR* genes.

## Figures and Tables

**Figure 1 plants-13-00668-f001:**
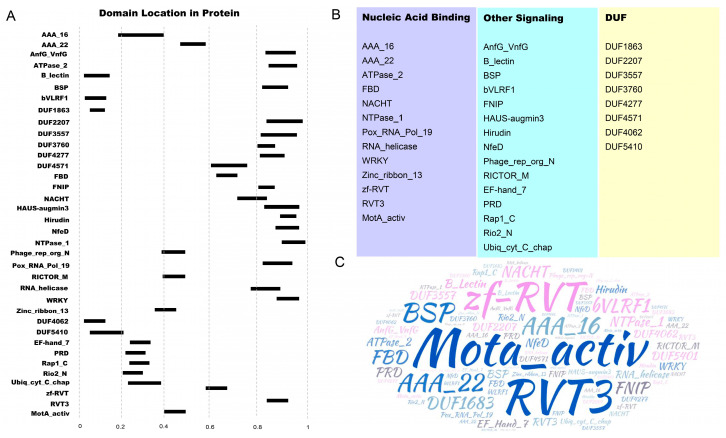
Overview of ID in soybean TNL-like proteins. (**A**) Integrated domain (ID) locations, indicated by black rectangles, are shown within soybean TNL-like proteins relative to protein length (0–1); (**B**) IDs were grouped into functional categories based on their potential involvement in nucleic acid-binding, other signaling activity pathways, or unknown functional pathways (shown in light purple, light blue, and light yellow, respectively); (**C**) word cloud analysis of the putative IDs found in fusion to soybean TNL-like proteins. The word cloud represents relative abundance of different domains found in fusion.

**Figure 2 plants-13-00668-f002:**
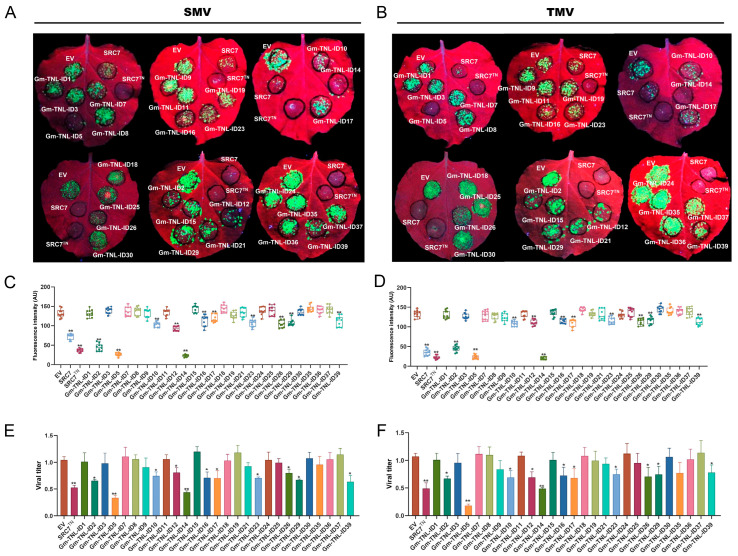
Antiviral activity of these *Gm-TNL-ID* genes in *Nicotiana benthamiana*. (**A**,**B**) Transient expression assay for antiviral activity. *N. benthamiana* leaves were infiltrated with *A. tumefaciens* inocula carrying different overexpression vectors for *Gm-TNL-ID* genes, and co-infected with SMV-GFP (**A**) or TMV-GFP (**B**). GFP was visualized under hand-held UV lamp (wavelength = 365 nm) at 4 dpi. SRC7 and SRC7^TN^: full-length SRC7 and truncations for SRC7 were used as a positive control. EV, empty vector. Each experiment was repeated three times with at least 20–30 technical replicates; (**C**,**D**) fluorescence intensity quantification of virus spread through Image J v1.8.0. The fluorescence diffusion of leaf infiltration sites under UV light after 4 dpi of mixed infiltration of SMV-GFP (**C**) and TMV-GFP (**D**), with the target gene was counted using software Image J v1.8.0 and analyzed for differences. *Y*-axis represents the mean fluorescence intensity of leaf infiltration region and *X*-axis represents the target gene. The fluorescence intensity of each gene was subjected to 10 technical replicates; (**E**,**F**) virus titer detection via RT-qPCR. Total RNA was extracted from a mixed sample of 10 leaves. *Y*-axis represents the mean virus titers of leaf infiltration region and *X*-axis represents the target gene. In *t*-test, bars represent SD of three independent biological replicates, each with three repeats, ns: no significant difference, *: *p <* 0.05, **: *p* < 0.01.

**Figure 3 plants-13-00668-f003:**
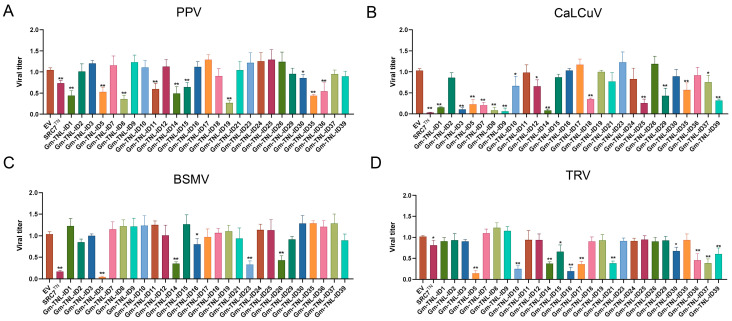
Identification of broad-spectrum resistance of *Gm-TNL-ID* genes in *Nicotiana benthamiana*. *N. benthamiana* leaves were infiltrated with *A. tumefaciens* inocula carrying different overexpression vectors and co-infected with PPV, CaLCuV, BSMV, or TRV. Tissue RNA was extracted from infiltrated region and was used for quantitative analysis of PPV (**A**), CaLCuV (**B**), BSMV (**C**), and TRV (**D**) virus titers. *Y*-axis represents the mean virus titers of leaf infiltration region and *X*-axis represents the target gene. In *t*-test, bars represent SD of three independent biological replicates, each with three repeats, ns: no significant difference, *: *p* < 0.05, **: *p* < 0.01.

**Figure 4 plants-13-00668-f004:**
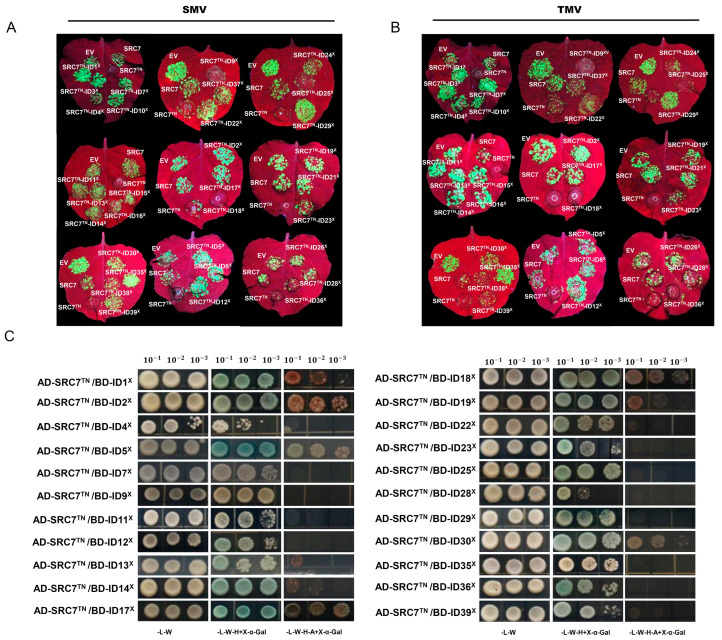
The effect of Gm-TNL-ID^X^ domain on antiviral resistance of SRC7^TN^ and its interaction with SRC7^TN^. (**A**,**B**) Transient expression assay for antiviral activity. *N. benthamiana* leaves were infiltrated with *A. tumefaciens* inocula carrying different overexpression vectors for SRC7^TN^-ID^X^ and co-infected with SMV-GFP (**A**) or TMV-GFP (**B**). GFP was visualized under hand-held UV lamp (wavelength = 365 nm) at 4 dpi. SRC7 and SRC7^TN^: full-length SRC7 and truncations for SRC7 were used as a positive control. EV, empty vector. Each experiment was repeated three times with at least 20–30 technical replicates; (**C**) Y2H assay. The yeast cells containing pGADT7 (AD) -SRC7^TN^ and different pGBKT7 (BD) -Gm-TNL-ID^X^ constructs were grown on different selective media, and 100 μL of X-α-Gal was added and visualized the reporter expression. The yeast concentration gradients were 10^−1^, 10^−2^ and 10^−3^, respectively. -L-T = yeast growth on medium lacking Leu and Trp, -L-W-H = yeast growth on medium lacking Leu, Trp, and His, -L-W-H-A = yeast growth on medium lacking Leu, Trp, His, and Ade.

**Figure 5 plants-13-00668-f005:**
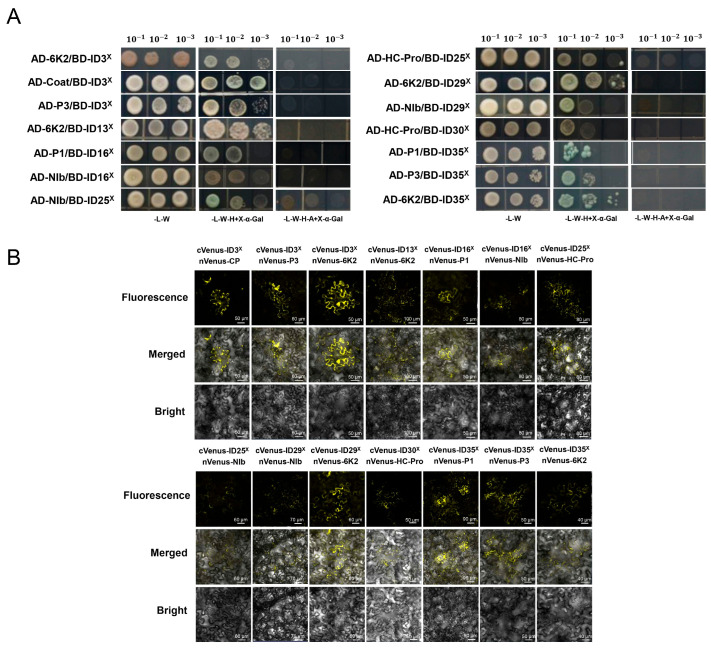
Interaction assay for Gm-TNL-ID^X^ domain and SMV-component proteins. (**A**) Y2H assay. The yeast cells containing ten AD-SMV-component proteins and different BD-Gm-TNL-ID^X^ constructs were grown on different selective media, and 100 μL of X-α-Gal was added and visualized the reporter expression. The yeast concentration gradients were 10^−1^, 10^−2^, and 10^−3^, respectively. (**B**) BiFC further validates the protein–protein interaction between the Gm-TNL-ID^X^ domain and SMV components in *N. benthamiana*. The Gm-TNL-ID^X^ and SMV-component proteins’ constructs were transiently expressed by co-infiltration of cVenus and nVenus fusions into *N. benthamiana* leaves. At 3 dpi, yellow fluorescent protein (YFP) fluorescence imaging was performed using confocal laser scanning microscopy.

**Figure 6 plants-13-00668-f006:**
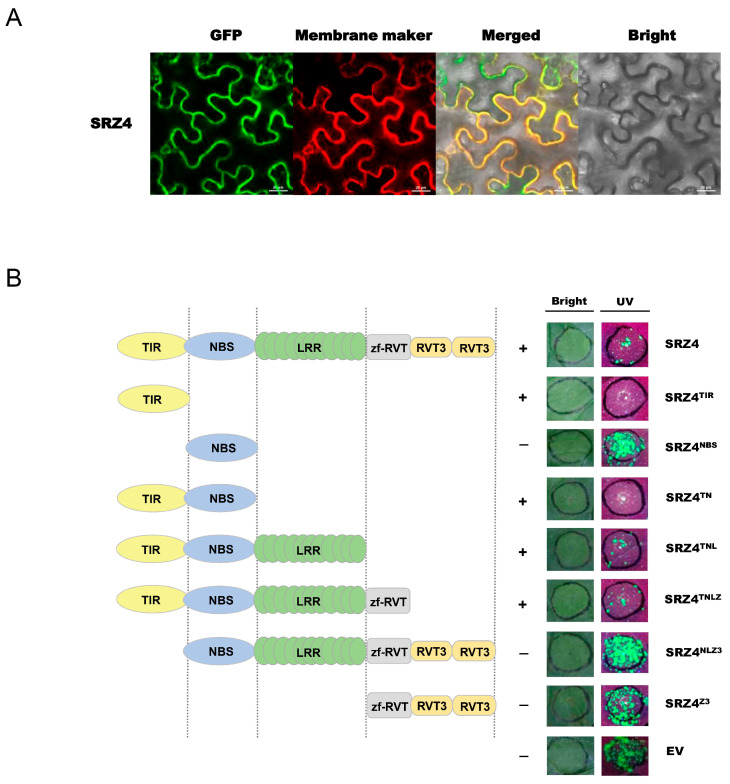
Subcellular localization and identification of antiviral functional domains of SRZ4. (**A**) Subcellular localization of SRZ4. The SRZ4–GFP fusion protein was transiently co-expressed with membrane marker (mCherry) in *N. benthamiana* leaf epidermis cells. The fluorescence signals were observed using confocal microscopy at 3 dpi. Scale bars = 20 µm; (**B**) transient expression assay for antiviral activity. *N. benthamiana* leaves were infiltrated with *A. tumefaciens* inocula carrying different SRZ4 truncated vectors, and co-infected with SMV-GFP. Each experiment was repeated three times with at least 20–30 technical replicates. GFP was visualized under hand-held UV lamp (wavelength = 365 nm) at 4 dpi. EV, empty vector.

**Figure 7 plants-13-00668-f007:**
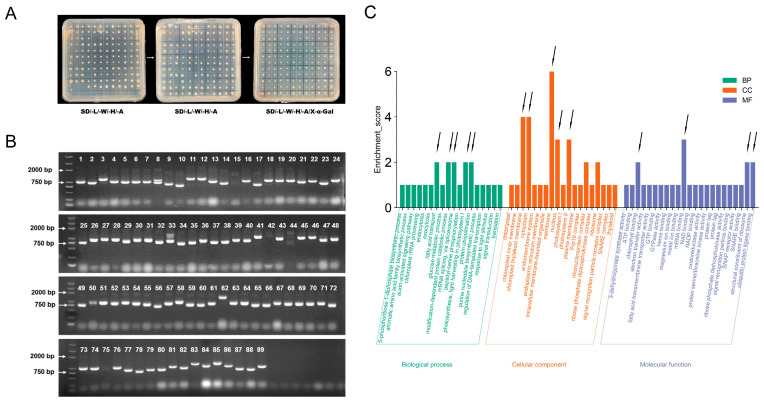
Interaction protein analysis of SRZ4. (**A**) Schematic diagram for screening and subculture of cDNA library. Using the mating method, BD-SRZ4 and soybean cDNA library were co-transformed into yeast cells and coated on 150 mm SD/-L/-W/-H plates. We selected colonies with good growth conditions and inoculated them onto SD/-L/-W/-H/-A plates. After 5–7 days, a total of 126 single colonies were obtained on SD/-L/-W/-H/-A plates. After three generations, selected single colonies that can turn blue and grow well in SD/-L/-W/-H/-A/X-α-Gal plate; (**B**) yeast colony PCR. PCR amplification was performed on each colony using universal primers from the yeast library, and PCR products with different band sizes were selected for sequencing; (**C**) GO enrichment analysis of SRZ4-interacting proteins. *X*-axis represents GO analysis annotation, where BP: biology process, CC: cellular component, and MF: molecular function. The *Y*-axis represents the number of SRZ4-interacting proteins. The black arrows represent the biological functions that interest us.

**Table 1 plants-13-00668-t001:** Most prevalent integrated domains in soybean.

Gene ID	Integrated Domain	Domain Description
*GLYMA_03G054100*	AAA_16	AAA ATPase domain
*GLYMA_12G240100*; *GLYMA_15G152400*	AAA_22	AAA domain
*GLYMA_06G310000*	AnfG_VnfG	Vanadium/alternative nitrogenase delta subunit
*GLYMA_16G087100*	ATPase_2	ATPase domain predominantly from Archaea
*GLYMA_06G261400*	B_lectin	D-mannose binding lectin
*GLYMA_16G214200*; *GLYMA_16G214300*	BSP	basic secretory protein
*GLYMA_01G112200*	bVLRF1	bacteroidetes VLRF1 release factor
*GLYMA_12G138400*; *GLYMA_12G138500*	DUF1863	MTH538 TIR-like domain (DUF1863)
*GLYMA_03G087500*	DUF2207	Predicted membrane protein (DUF2207)
*GLYMA_09G056400*	DUF3557	Domain of unknown function (DUF3557)
*GLYMA_06G265400*	DUF3760	Protein of unknown function (DUF3760)
*GLYMA_06G265000*	DUF4277	Domain of unknown function (DUF4277)
*GLYMA_13G194700*	DUF4571	Domain of unknown function (DUF4571)
*GLYMA_13G078200*	FBD	FBD
*GLYMA_15G152200*	FNIP	FNIP Repeat
*GLYMA_14G151100*; *GLYMA_16G085400*	NACHT	NACHT domain
*GLYMA_01G032400*	HAUS-augmin3	HAUS augmin-like complex subunit 3
*GLYMA_03G052800*	Hirudin	Hirudin
*GLYMA_03G077400*	NfeD	NfeD-like C-terminal, partner-binding
*GLYMA_07G037000*; *GLYMA_16G006400*	NTPase_1	NTPase
*GLYMA_03G047900*	Phage_rep_org_N	N-terminal phage replisome organiser
*GLYMA_16G085700*	Pox_RNA_Pol_19	Poxvirus DNA-directed RNA polymerase 19 kDa subunit
*GLYMA_07G123000*	RICTOR_M	Rapamycin-insensitive companion of mTOR, middle domain
*GLYMA_03G075300*	RNA_helicase	RNA helicase
*GLYMA_05G165800*	WRKY	WRKY DNA -binding domain
*GLYMA_06G285500*	Zinc_ribbon_13	Nucleic-acid-binding protein containing Zn-ribbon domain
*GLYMA_12G221600*	DUF4062	Domain of unknown function (DUF4062)
*GLYMA_02G023800*	DUF5410	Family of unknown function (DUF5410)
*GLYMA_16G214500*; *GLYMA_16G214800*	EF-hand_7	EF-hand domain pair
*GLYMA_02G023900*	PRD	PRD domain
*GLYMA_06G146200*	Rap1_C	TRF2-interacting telomeric protein/Rap1-C terminal domain
*GLYMA_06G268500*	Rio2_N	Rio2, N-terminal
*GLYMA_02G023700*; *GLYMA_02G024000*	Ubiq_cyt_C_chap	Ubiquinol-cytochrome C chaperone
*GLYMA_16G210600*; *GLYMA_16G210800*; *GLYMA_16G127900*; *GLYMA_09G161400*; *GLYMA_08G301200*	zf-RVT, RVT3	zinc-binding in reverse transcriptase
*GLYMA_16G135500*; *GLYMA_16G136200*; *GLYMA_16G135200*; *GLYMA_16G137300*; *GLYMA_16G136000*; *GLYMA_16G147400*; *GLYMA_19G054700*; *GLYMA_19G054900*; *GLYMA_16G159600*; *GLYMA_19G022700*; *GLYMA_16G213700*; *GLYMA_06G267300*; *GLYMA_06G268600*	MotA_activ	Transcription factor MotA, activation domain

**Table 2 plants-13-00668-t002:** Seven Gm-TNL-ID^X^ domains interacting with SMV-component proteins.

Gene Names	Gene ID	Integrated Domain	Length (aa)	Interaction Protein
*Gm-TNL-ID3*	*GLYMA_03G087500*	DUF2207	639	CP, P3, 6K2
*Gm-TNL-ID* *13*	*GLYMA_09G161400*	zf-RVT	957	6K2
*Gm-TNL-ID* *16*	*GLYMA_06G285500*	Zinc ribbon 13	665	P1, NIb
*Gm-TNL-ID* *25*	*GLYMA_06G267300*	MotA activ	195	NIb, HC-Pro,
*Gm-TNL-ID* *29*	*GLYMA_19G054900*	MotA activ	704	6K2, NIb
*Gm-TNL-ID* *30*	*GLYMA_19G054700*	MotA activ	656	HC-Pro
*Gm-TNL-ID* *35*	*GLYMA_16G135500*	MotA activ	633	P1, P3, 6K2

## Data Availability

The data used in this research are publicly available. The protein sequences of each cloned gene can be found at https://plants.ensembl.org/index.html (accessed on 20 December 2020) and https://www.ncbi.nlm.nih.gov/ (accessed on 20 December 2020). The data (results) presented in this research are available in the Appendix A.

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
