# Peer review of "Development of an NLR-ID Toolkit and Identification of Novel Disease-Resistance Genes in Soybean"

_plants, 2024, doi:10.3390/plants13050668_

Round 1

Reviewer 1 Report

Comments and Suggestions for Authors

Dear Authors,

The Introduction provides a good overview of the background of this study.

The methodology is presented in detail and consistently. The experimental design is appropriate and the manuscript is well-structured.

 Results and discussion described reliably and clearly.

 Therefore, the current draft does not need to be revised before further process.

Reviewer 2 Report

Comments and Suggestions for Authors

Shao et al. screened soybean genome for the identification of TNL-like proteins with atypical integrated domains (IDs), and experimentally tested the functions of 27 Gm-TNL-ID genes. Among these genes, SRA2 and SRZ4 are SMV resistance genes, and they exhibit broad-spectrum resistance against to TMV, PPV, CaLCuV, BSMV, and TRV too;The  SRZ4TIR is sufficient to mediate antiviral activity against SMV. Additionally, the ID domains of most Gm-TNL-ID inhibit antiviral activity mediated by SRC7TN. These results are suitable for publishing in Plants. However, two issues below need to be addressed before publication:

1. Too many abbreviations are used in the abstract without giving their full names, which very likely will makes readers confusing. It is probably a good idea to reduce the number of abbreviations and replace them with descriptive words.

2. Although six genes interact with SRZ4 are identified, it is not clear how the interactions affect the antiviral activity of SRZ4. Thus the results are irrelevant to the manuscript. My suggestion is that not including these results in the manuscript, which would help to shorten the manuscript as well. A manuscript of 24 pages is very long.

Comments on the Quality of English Language

The overall quality of English is pretty good.

Reviewer 3 Report

Comments and Suggestions for Authors

Abstract

Improve a little.

Introduction

Write about this species, breeding, origin, chromosome numbers, history.

Add some fresh research on it.

Materials and methods

How about control, home you wanted to compare,

what was the method of getting samples, you mixed samples for RNA isolation or separate them for each seedling?

How about linkage genes to chromosomes/position etc.

for clear understanding control is necessary.

Results

Explain with samples, give more detail where possible.

Co-relate the genes with parameters/standard of genes.

Discussion

give little detail regarding your samples

Reference:

add references according to format of journal. 

Comments on the Quality of English Language

Improve english after corrections
